# A Review on the Research Advances in Groundwater–Surface Water Interaction with an Overview of the Phenomenon

Dolon Banerjee and Sayantan Ganguly *

Department of Civil Engineering, Indian Institute of Technology Ropar, Rupnagar 140001, Punjab, India; dolon.20cez0004@iitrpr.ac.in
* Correspondence: sayantan.ganguly@iitrpr.ac.in or sayforall@gmail.com

**Abstract:** Groundwater and surface water, though thought to be different entities in the past, are connected throughout the different landforms of the world. Despite being studied for quite some time, the interaction between groundwater and surface water (GW–SW) has received attention recently because of the heavy exploitation of both of these resources. This interaction is responsible for a phenomenon like contaminant transport, and understanding it helps to estimate the effects of climate change, land use on chemical behavior, and the nature of water. Hence, knowledge of GW–SW interactions is required for hydrologists to optimize resources and analyze the related processes. In this review article, different aspects of the interaction are discussed. Starting from the basics of the phenomenon, this work highlights the importance of GW–SW interactions in the hydrological cycle. Different mechanisms of GW–SW interactions are briefly examined to describe the phenomenon. The scales of interaction are also elucidated where the classification is addressed along with a brief introduction to the large scale and sediment reach scales. The study then moves on to the investigation methodologies used for the process of SW–GW interaction and their classifications based on whether they are field methods or modeling techniques. Various literature is then explored in terms of research approaches. Finally, we highlight the applicability of the methods for different scenarios. This work is aimed to summarize advances made in the field, finding research gaps and suggest the way forward, which would be helpful for hydrologists, policymakers and practicing engineers for planning water resources development and management.

**Keywords:** groundwater–surface water interaction; gaining and losing streams; Darcy approach; hydrochemistry; hydrological cycle; water budget

## 1. Introduction

The interaction of the two important parts of the water cycle, Groundwater (GW) and Surface water (SW), were considered to be different entities in the past and were examined and quantified separately for a long time. With time, their profound interdependency has been explored. The interaction of GW–SW takes place in various ways in all landscapes of the earth [1]. The interaction phenomenon commences as the water enters the hyporheic zone from either of the sources. The term hyporheic is derived from Greek roots—hypo, meaning under or beneath, and rheos, meaning a stream (rheo means to flow). Valett [2] describes the hyporheic zone as the region below streams and rivers that exchanges water with the surface sources, whereas Triska [3] defined this zone as the part beneath the surface water body containing contributions both from surface water and groundwater, but has surface water greater than 10 percent of the total volume. The hyporheic zone contains high levels of organic carbon and microbes, facilitating the breakdown of pollutants from the surface or groundwater into simpler and harmless byproducts. This interaction between water, nutrients, and biodegrading organisms occurs via bio-films and is influenced by sediment quality and properties, affecting the residence time. The hyporheic zone also alters the chemical composition of incoming water and plays a crucial role in contaminant transport and stream processes.

The classification of surface water-aquifer systems is based on the degree of interaction between them, with six different types identified [4]. A gaining stream (Figure 1a) occurs when groundwater seeps into the stream, while for a losing stream (Figure 1c,d), water seeps from the stream into the aquifer. Transition-losing streams (Figure 1b), on the other hand, experience both sorts of interactions. Hydraulically disconnected streams have a thick unsaturated zone between the stream and groundwater, while losing and parallel connected streams (Figure 1e) have the groundwater table at or below the stream bed. Flow through streams (Figure 1f) have differing groundwater levels on either side of the stream bed. Groundwater and surface water are linked, and their interactions affect the hydrologic cycle and human life. Extraction and pollution can harm both systems, making it crucial to understand the interconnections for effective land and water management. Progress in research has emphasized quantitative and qualitative estimation of surface water-groundwater interactions to analyze phenomena in the riparian zone. In the 1960s, the GW–SW interaction between lakes and groundwater was studied to understand acid rain and eutrophication [5]. Similarly, from the 1960s through the 1980s, researchers focused more on the interaction between groundwater and wetlands, and coastal areas because the ecosystems involved were on the verge of extinction [6].

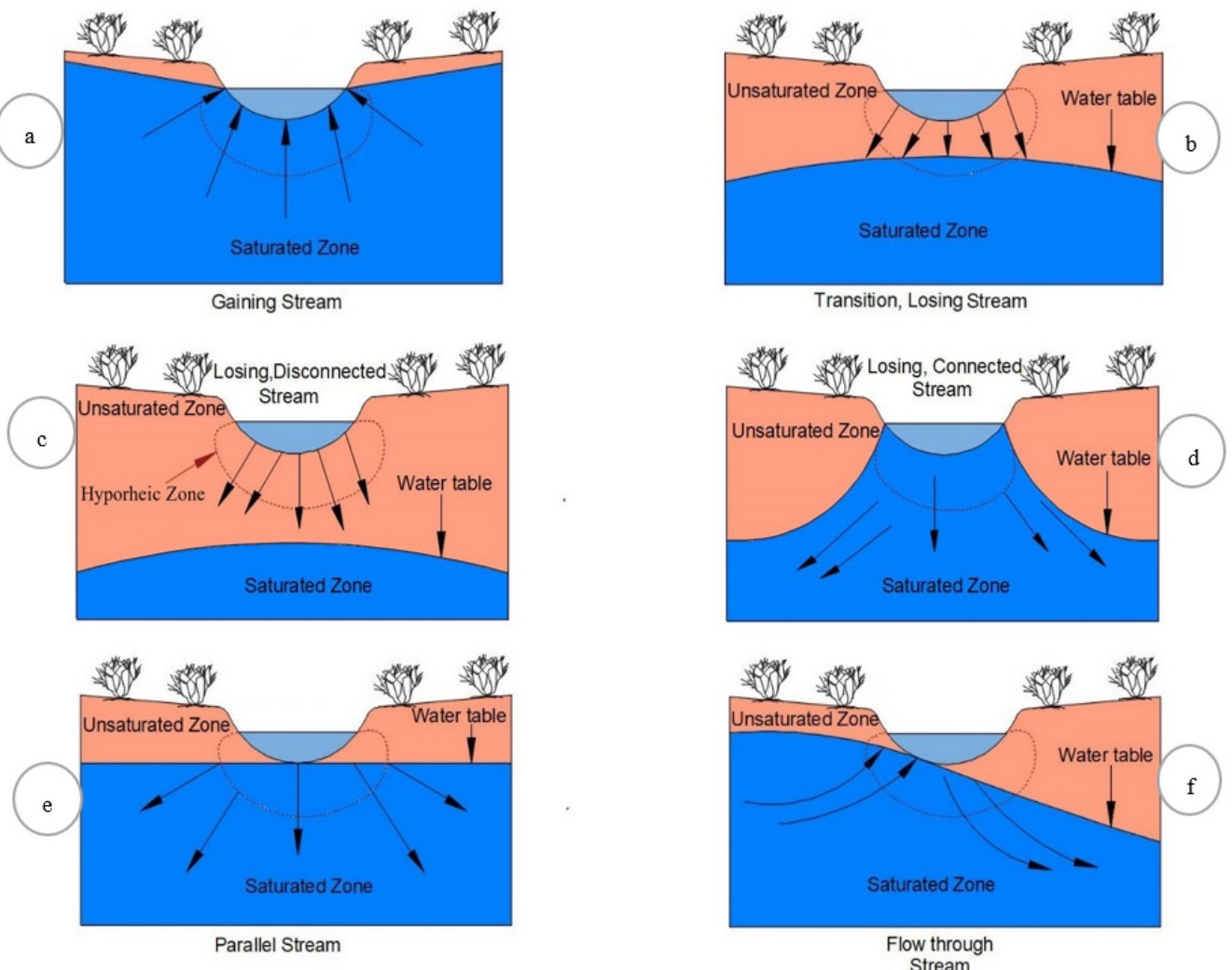

**Figure 1.** Different stream-water and groundwater interaction scenarios, (**a**) gaining stream, (**b**) transition-losing stream, (**c**) losing-disconnected stream, (**d**) losing-connected stream, (**e**) parallel stream, (**f**) flow-through stream. The arrows denote the directions of fluid flow.

Around the mid-1950s, in several places around the globe, groundwater pumping was found to influence the in-stream flows [7–9]. Seepage flux measurement in lakes and estuaries was done using a seepage meter and mini piezometers, which helped to understand the interaction of streamflow to groundwater quantitatively [10,11]. The variation of surface and subsurface water exchange over different seasons along the hyporheic area of two stream-aquifer systems was evaluated to address the variation in stream discharge and groundwater level [12]. Later, due to the increasing interest in ecological and climatic concerns, the GW–SW interaction along a river's hyporheic zone got researchers' attention [5,9]. Over time, many different methods have been developed to accomplish this task, ranging from simple continuity equations to complex modeling techniques [1,5]. One of the most straightforward ways to measure water flux and estimate GW–SW interaction is by using a seepage meter to measure water flow [11]. Heat tracers can also be utilized to determine water flux and delineate recharge zones by measuring the temperature difference between GW and SW [13]. Another popular method is the mass balance approach, which posits that any changes in the volume of a surface water body are related to its interaction with surrounding groundwater. This approach allows for the calculation of the flow between GW and SW and the linking of surface water attributes to their water source. Darcy's Law is a highly effective tool that can track and quantify GW's movement through soil and its addition to and from surface water [14]. Negral [15] used a combined approach to study GW–SW interaction in transitional wetlands, considering hydrological, geochemical, ecological, and sociological aspects. The challenge is to quantify flux and understand its spatial and temporal variation [13]. Isotope readings were used to determine if groundwater was being recharged by local rainfall and surface water sources or was recharging the river as baseflow in a catchment [14]. Grodzka-Łukaszewska et al. [16] studied GW–SW interaction in Poland using two measurement campaigns and a groundwater flow model. They measured flux, infiltration flux density, and drainage density using a seepage meter, filtrometer, and gradient meter. The model was verified using measurement data and showed a good correlation between observations and results. Grodzka-Łukaszewska [17] studied GW–SW interaction in the Biebrza River and its impact on peat habitats. They used FEEFLOW software to model interaction and measured piezometric readings and pressure differences with gradient meters. A water balance approach was used to analyze processes. Results showed that the river has a draining character and contributes only 10% to peat layer recharge. Anibas et al. [18] developed a hierarchical approach to analyze GW–SW interaction using piezometer nests, temperature tracers, and seepage meters. They used STRIVE, a 1-D heat transport model, to calculate vertical exchange fluxes at the Biebrza River. Results revealed upward water fluxes with recharge sections along the reach.

Research on groundwater and surface water interaction involves interdisciplinary issues such as the use of geophysical techniques. Geophysical methods can provide information on subsurface properties such as geological, hydrological, and biogeochemical properties [19]. These methods include electrical resistivity, induced polarization, self-potential, electromagnetic induction, groundwater penetrating radar, and various seismic methods. They are helpful in determining water content, subsurface composition, clay content, permeability, and conductivity. Electrical resistivity and seismic methods can accurately determine the porosity and stratigraphy of the sub-surface [20]. The results obtained from these methods are interpreted through petrophysical models, temporal data analysis, and calibration with other methodologies, along with the most common forward and inverse modeling techniques [21]. However, there are a number of challenges like geophysical uncertainty, site-specific considerations, modifications, and the need for good and in-depth knowledge for processing and modeling the collected results to get the final quantitative interpretation. Groundwater exchange is also crucial for maintaining the ecological balance of ecosystems such as rivers, streams, and lakes [22]. This exchange influences the ecology of surface water bodies both directly and indirectly. In streams, it sustains the base flow, and in lakes, it moderates water-level fluctuations. The interaction also regulates temperature in the hyporheic zone and helps biota survive through seasonal

variations. Groundwater and surface water supply nutrients and inorganic ions to each other [23,24].

This paper traces the development of methods for investigating SW–GW interaction over time, focusing on different estimation methods and their applicability as found in a wide range of literature. The paper discusses numerical, analytical, and semi-analytic methods used for groundwater and surface water interaction, as well as geophysical methods for quantification. Recent literature is also reviewed to show how different estimation methods are applied to complex problems. Furthermore, the outline, along with their arrangement in the paper, is shown in Figure 2. The paper presents the phenomenon's significance and factors and discusses its scale-dependent variation and analysis methods. Literature review and method applicability over a wide range of scenarios are also discussed, followed by a conclusion and suggestions for future research.

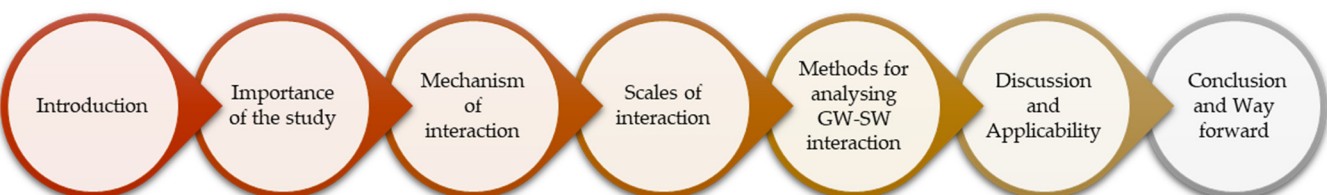

**Figure 2.** Layout of the paper.

## 2. Importance of SW–GW Interaction

Surface water and groundwater contribute to each other as a source and sometimes as a sink. The contribution of groundwater to oceans, streams, and lakes was also quantified [25]. They reported groundwater and surface water exhibit a profound interaction, with groundwater contributing almost 6% of freshwater fluxes to oceans and 35% to 55% of stream runoff. A study on 24 regions in the USA found that groundwater contributed to surface water between 14% to 90%, with a mean of 55% [1]. Later, a study demonstrated that 70% of submarine groundwater discharge flows into the Indian and Pacific oceans, unlike rivers which discharge almost half the total flux into the Atlantic Ocean [26]. The profound connection between groundwater and lakes in North America was found as groundwater nearly contributed 0% to 94% to the lakes, and lakes, too, had a contribution of 0% to 91% to the groundwater [27]. Hence, knowing about this GW–SW interaction helps us to understand their nature and extent of involvement with each other for planning water resources management. Agricultural activities, septic systems, and sewers can contaminate groundwater, which then contaminates streams and lakes through baseflow. This contamination typically includes high nitrate levels and minor contents of many other nutrients [28]. Groundwater has higher dissolved solids than surface water, which can result in the transfer of nutrients and salts to surface water resources. This has been demonstrated in Adirondack lakes in the US, which had higher base cations and metals seeping through groundwater, leading to eutrophication [29,30].

Surface water sources can also contaminate groundwater in several cases. A study in Chennai, India, reported that high concentrations of toxic elements in the groundwater were found in areas where surface water was heavily contaminated with toxic elements [31]. Bear studied the intrusion of ocean water and salts into groundwater, which can lead to the contamination of other surface water bodies [32,33]. Singh found that heavy metals, as well as calcium, sulfur, and nickel, were present in higher concentrations in groundwater near the Buddha Nullah River, Ludhiana, India, with high TDS and BOD levels [34]. Maeng showed that organic micropollutants from pharmaceuticals can deteriorate water quality in areas where they are discharged, with further effect on supply water quality [35]. Li found that anthropogenic ions ($Na^+$, $Cl^-$, $NO_3^-$) and nutrients intrude into groundwater along the Fenhe River in the Jinci karst system in China [36]. Prakash found higher concentrations of trace elements like Al, Cr, Fe, Pb, and Zn near the Bay of Bengal in India than away from it [37]. Guevara-Ochoa [38] demonstrated that climate change can

modify groundwater levels and reverse GW–SW flow in some reaches of streams, causing variations on a monthly, seasonal, or annual basis. Abdelhalim [39] found similar results with experiments on the river Nile, showing that climate change decreases both surface water and groundwater levels.

## 3. Mechanisms of GW–SW Interaction

Surface and subsurface water interact through water infiltration from the surface to the subsurface water table or exfiltration from the saturated zones, as well as the lateral flow of water in the subsurface zone that emerges into a surface water body. Sophocleous [5] demonstrated how karst terrain has these interactions occurring through flow in fracture channels. For a general soil profile, Beven [40] identified four mechanisms by which subsurface flow contributes to streamflow in a brief period, in addition to surface runoff from a single rainstorm input. The mechanisms are: (a) translatory flow, (b) macropore flow, (c) groundwater ridging, and (d) return flows.

Translatory flow, also known as plug flow or piston flow, is a lateral flow in which the water stored in the voids of soil structure before the storm is displaced by the percolated rainfall water, hence forming a component of subsurface storm flow. It may be called lateral flow if old water is displaced by precipitation input. Translatory flow in a lab is simulated by taking a soil column, letting it drain to field capacity, and adding water at the top [41,42].

Macropore flow is the type of flow in which there is a continuous flow from the soil surface to the groundwater table, not getting trapped or losing water in the intermediate soil profile. This flow occurs through connected and disconnected macropores, soil pipes, soil cracks, random holes formed by soil fauna, and desiccated roots [43]. Macropore flow consists of 'old' or 'pre-event' water, which has a quick subsurface contribution. When the water flow under pressure greater than or equal to atmospheric pressure, which means either water is inside the saturated zone or there is a ponding state at the surface of the earth, it enters a large non-capillary pore [44].

The third phenomenon is groundwater ridging, in which the rapid increase of hydraulic head near the stream causes a substantial contribution from groundwater to the stream. Above the groundwater table exists a capillary fringe zone with water held under surface tension. During a storm, this fringe gets destroyed just by adding a small amount of water into this zone, so the water rises to the top of the fringe. In this process, water pressure inverts from negative to positive. Due to the water level rise near the stream, the net hydraulic gradient increases or the seepage face causing more significant groundwater discharge to the stream, and thus induced discharge from the groundwater to the stream may be higher in quantity than that the input water that triggered the process [5].

The fourth phenomenon, Return Flow, is an extension of Groundwater Ridging, which occurs when the water table and capillary fringe are very near to the soil surface, and even a minimal amount of percolated water will cause the capillary to break. Hence pressure inverts from negative to positive with the water table rise. Still, this saturated soil will start discharging water from the subsurface to the surface directly, which is termed Return Flow. According to Beven [40], the contribution area of return flow depends upon the closeness of the capillary fringe to the surface. It shows expansion if this area is close to the surface.

Apart from the mechanism suggested by Beven [40], another predominant phenomenon for the interaction is Induced Riverbank Flow. When water is pumped from a well, it creates a pressure gradient that induces flow from the river to the well, which leads to groundwater recharge. This induced recharge process enhances the interaction between the river water and groundwater, affecting the hydrodynamics and chemistry of both systems. Understanding the role of induced riverbank flow is important for designing and operating riverbank filtration systems that rely on this interaction to provide safe and reliable drinking water [45]. The same concept has been explored by Rossetto et al. [46]. They have used multidisciplinary methods like hydrodynamics, hydrochemical, and numerical modeling to evaluate the change in recharge from the Serchio River to the aquifer due to the building of the Riber Bank Filtration infrastructures along the river. They

established that the pumping wells alongside the river are being fed through the river and that the use of induced recharge would drastically increase the river water level up to 1.5 m. Zu et al. [47] studied the water supply safety of riverbank filtration wells under the impact of surface water-groundwater interaction. They have also shown that long-term pumping may impact the efficiency of riverbank filtration wells.

**4. Scales of GW–SW Interactions**

Tóth [48] introduced the term Groundwater Flow Systems for the classification of groundwater, which is a set of aquifers having similar characteristics that exhibit a definite pattern to the flow of water through them (Figure 3). For an area of a few hundred square kilometers with a mild slope and lower-order outlet stream, Tripathi [49] divided the flow scales into local, intermediate, and regional for unconfined groundwater systems. Winter [1] demonstrated that similar flow systems classification is effectively applicable to groundwater systems with confined aquifers. Tóth [48] stated that the scales that come into the picture for a particular case depend upon local and regional geomorphology. Local flow systems depend upon the local slope of an area and diminish or even get extinct if the regional slope is increased, which caters to the formation of other flow systems. In local systems, discharge fluctuates widely, and water flux has higher penetration depth and residence. The hydrologic properties also change according to the scale of the flow. Groundwater and surface water interaction highly depend on the scale through which their interaction occurs [50].

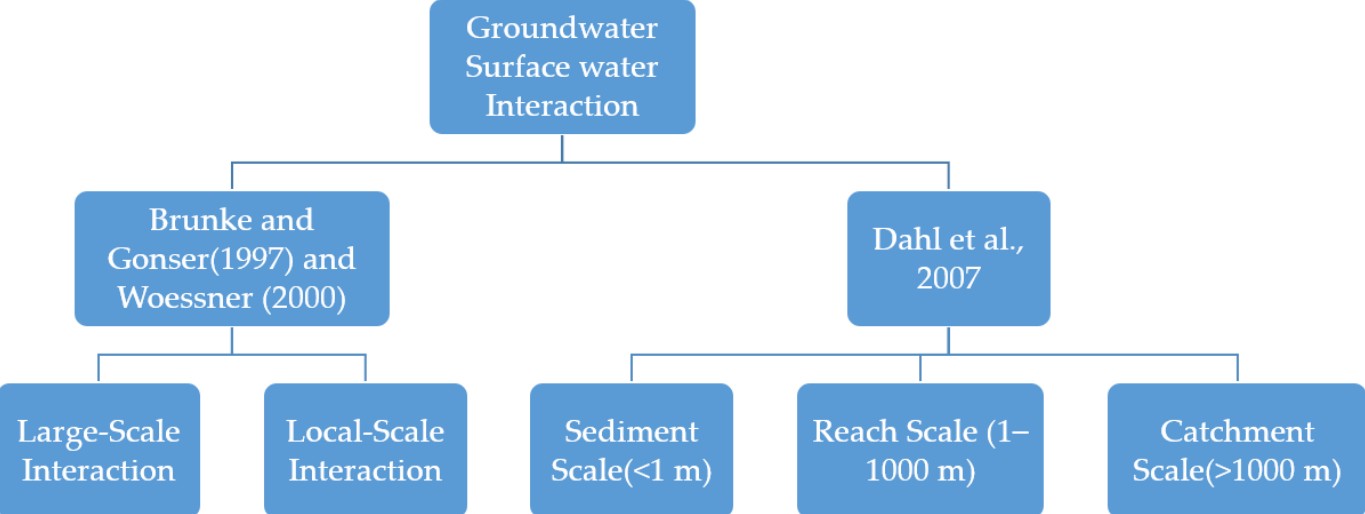

**Figure 3.** Different groundwater–surface water interaction scales studied through different approaches [50–52].

As shown in Figure 4, the interactions were distributed into different sets according to the scale of their interaction. The scales of GW–SW interactions were separated into two types [51,52], namely large-scale and local-scale interactions. Large-scale interaction was used when the whole of the catchment was actively participating in the interaction process, whereas local-scale interaction was used when only the hyporheic zone was influencing the interaction process.

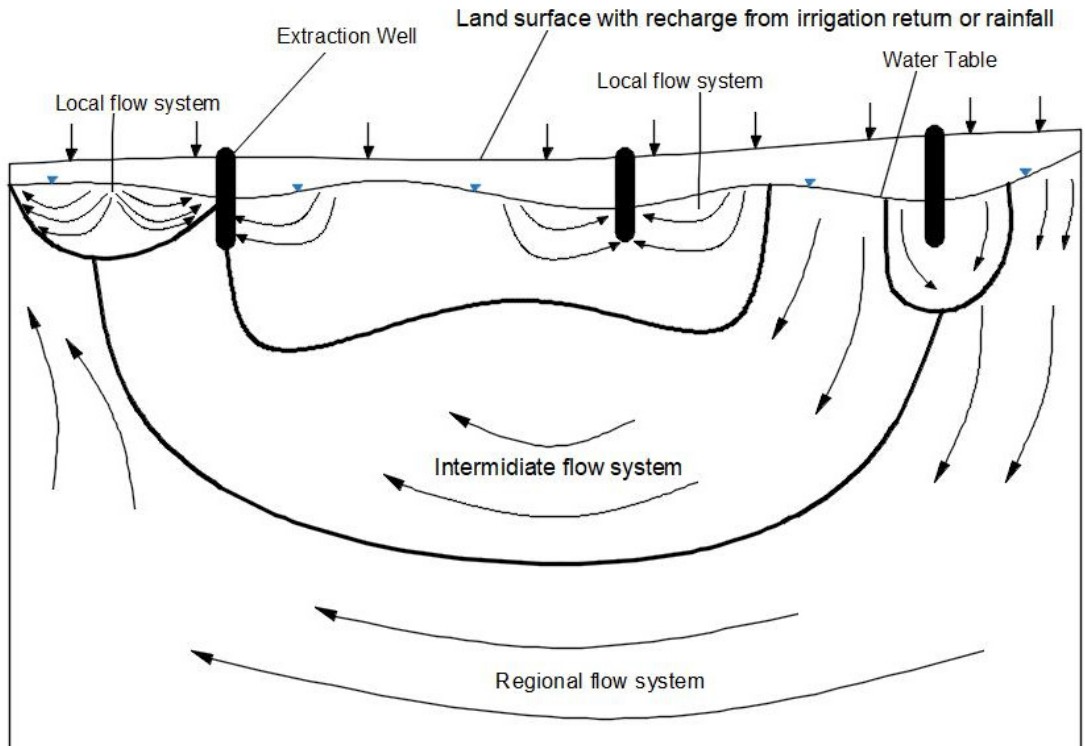

**Figure 4.** Groundwater flow systems: Local, Intermediate and Regional, as described by Tóth [48].

Dahl [50] divided the groundwater and surface water interaction into three sub-areas according to the scales, the first being the sediment zone within 1 m depth, the reach zone covers a depth of up to 1000 m, and a catchment zone is concerned with a depth of more than 1000 m. These scale divisions further resemble the hierarchic classification of groundwater flow systems. The hyporheic zone correlates to the sediment scale, whereas the local flow system corresponds to the reach size and the regional flow system to the catchment scale. The most commonly encountered interaction scales are given below:

(a)     Large-scale Interaction

On a regional or local scale, the interaction between groundwater and surface water depends on the position of the water body respective to groundwater flow systems, anisotropy of the soil system underneath and hydraulic conductivity variations of the groundwater system, arrangement of the water table, and depth of concerned water body.

Groundwater flow depends on the water table elevation relative to surface water-bed elevation. However, it has been observed that sometimes, even with a higher water table elevation, surface water discharges water to groundwater. The local groundwater flow system boundary controls these processes. Winter [53] suggests that seepage through a streambed occurs when there is no continuous local groundwater flow system boundary or stagnation point under the surface water body. There is no seepage if there is a continuous local groundwater flow system boundary or a stagnation point (Figure 5). The head difference between the surface water body and a stagnation point determines the amount of seepage.

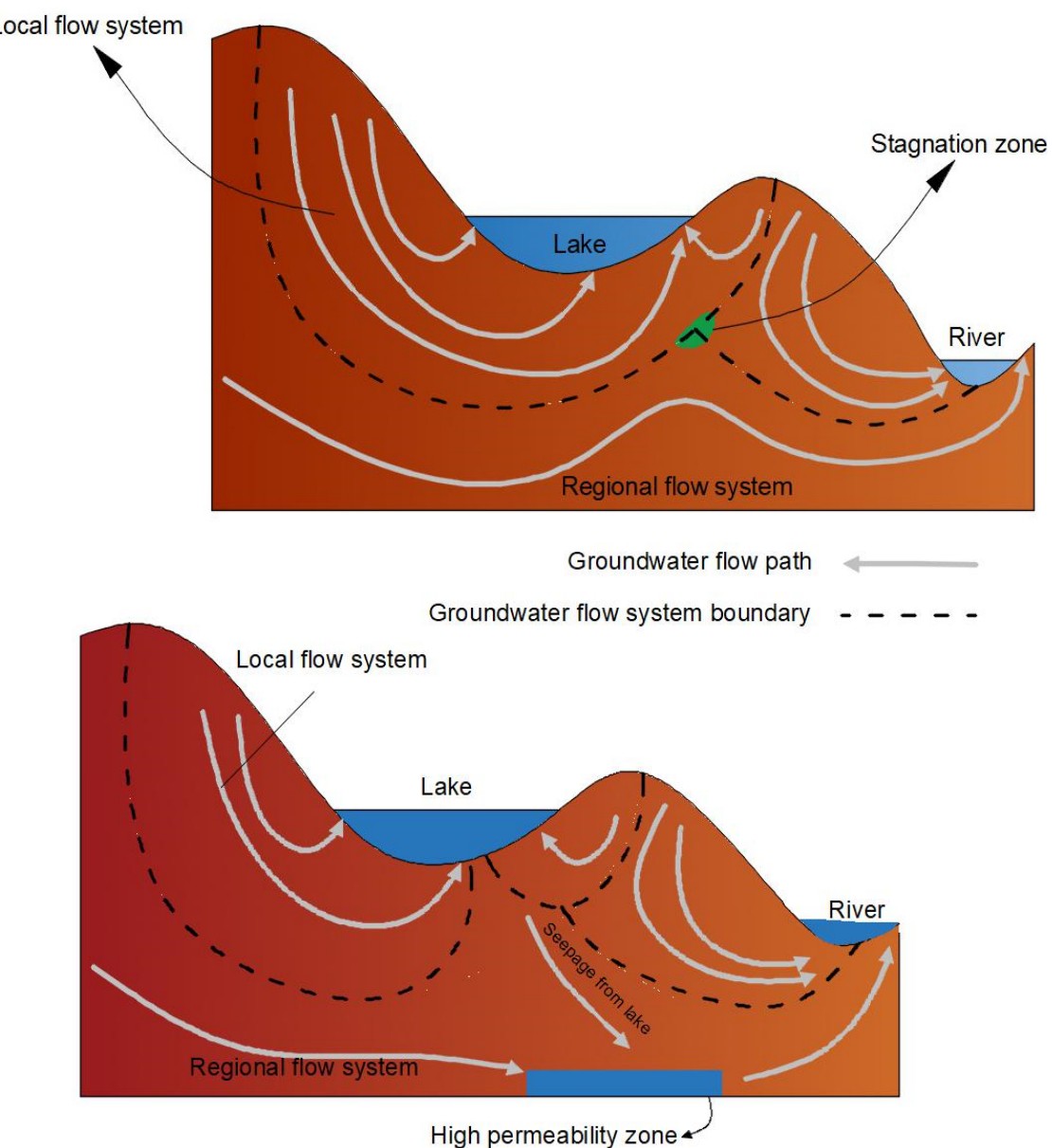

**Figure 5.** Conditions for seepage to occur from a surface water body with respect to groundwater flow system boundary.

(b) Hyporheic interaction or sediment scale interaction

The interaction between the surface water bodies and the water stored in the sediment directly underneath the water bodies is termed the Hyporheic Interaction (Figure 6). This interaction accounts for the local water infiltration from the streambeds and stream sides to the aquifer underneath and vice versa. In addition, a stream may have localized zones of infiltration and exfiltration even though the overall effect may be reversed [51,52]. According to Woessner [51], the highly localized flow systems are mainly controlled by surface-water-bed topology and sediment hydraulic conductivity variation beneath the stream bed.

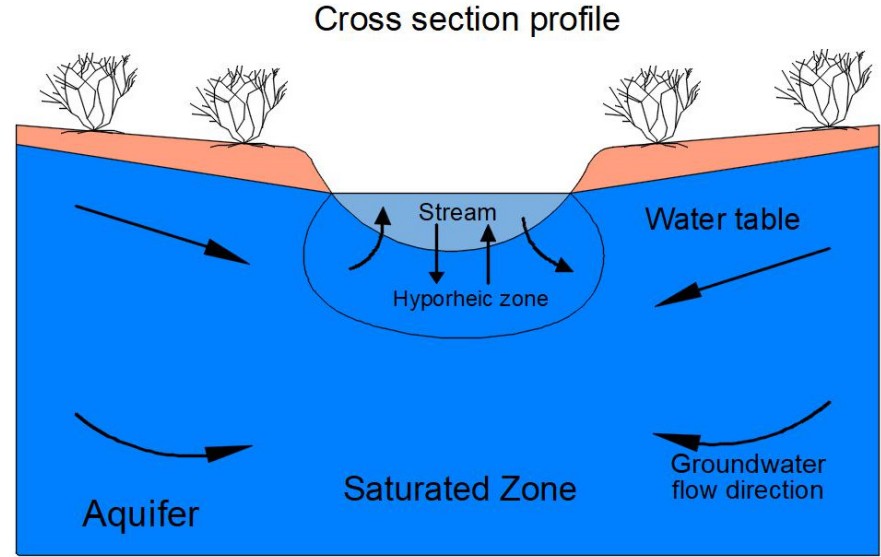

**Figure 6.** Processes occurring in the Hyporheic Zone. The black arrows represent the fluid flow directions.

Harvey and Bencala [54] showed that the interaction between a stream and underlying sediments is influenced by bed convexity and concavity. Due to the stream bed convexity, downwelling of the stream occurs while upwelling of the hyporheic and deep waters occurs due to concavity. The water enters through the riffles, the convex part, and exits through the pool area, the concave part of the stream bed. Cardenas [55] found that stream water enters the deposits through the upstream portion of a meander and moves back to the stream through the downstream portion of the meander, influenced by the factor channel sinuosity. Woessner [51] demonstrated that the hydraulic conductivity of stream bed sediments affects the depth of mixing of surface water and groundwater, with heterogeneous bed sediments increasing the depth of mixing to 1.5 m underneath the streambed compared to 0.7 m with homogenous bed sediment.

## 5. Methods for Analyzing GW–SW Interaction

We now go through the estimation methods of groundwater and surface water interaction. We will discuss estimation methods under four broad headings (Figure 7).

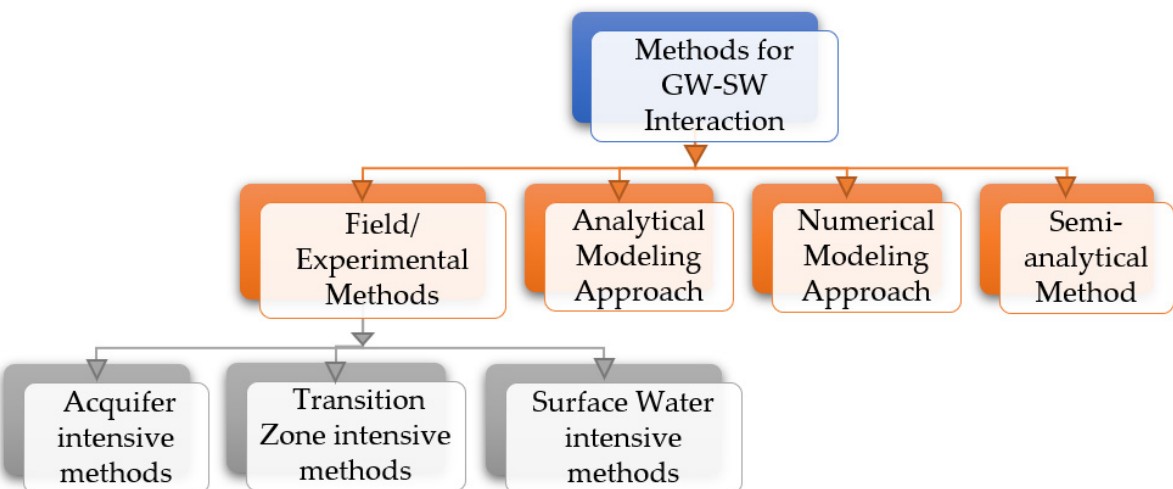

**Figure 7.** Methods for analyzing groundwater (GW) and surface water (SW) interactions.

### 5.1. Field/Experimental Methods

#### 5.1.1. Aquifer Intensive Methods

To understand groundwater–surface water interaction, it's important to understand the flow of groundwater in relation to surface water and topography. Water table elevation maps can help determine the direction of groundwater flow and identify gaining and losing stream reaches by analyzing contour lines. Increasing contour lines indicate groundwater contributing to the stream, while decreasing contour lines indicate the stream contributing to groundwater.

The water table map can be used to understand the phenomenon taking place, but for quantification purposes, in this case, Darcy's Law comes in handy. For the groundwater and surface water interaction, we are concerned with seepage velocity or groundwater velocity, so the Darcy equation is rewritten as:

$$v = \frac{q}{n} \tag{1}$$

where $v$ is groundwater velocity [L/T], $q$ is Darcy flux [L$^3$/T], and $n$ is porosity. Hence, hydraulic gradient, hydraulic conductivity, groundwater velocity, and porosity are required to be quantified to obtain the water flux in the subsurface (Figure 8).

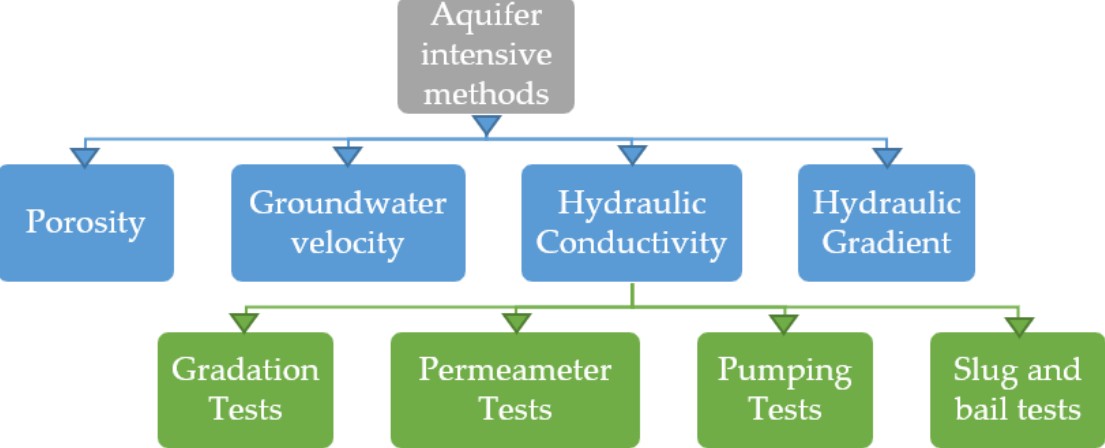

**Figure 8.** Aquifer intensive methods for quantifying GW–SW interactions.

#### 5.1.2. Surface Water Intensive Methods

To study the GW–SW interactions with respect to the surface water sources, we employ the mass balance approaches, which work on the assumption that any physical or properties-related change undergone by the stream is a reflection of some change in the corresponding water source only. Hence, we can identify and quantify the groundwater component. Some of the methods for quantification of groundwater interaction with the surface flow are discussed herewith (Figure 9).

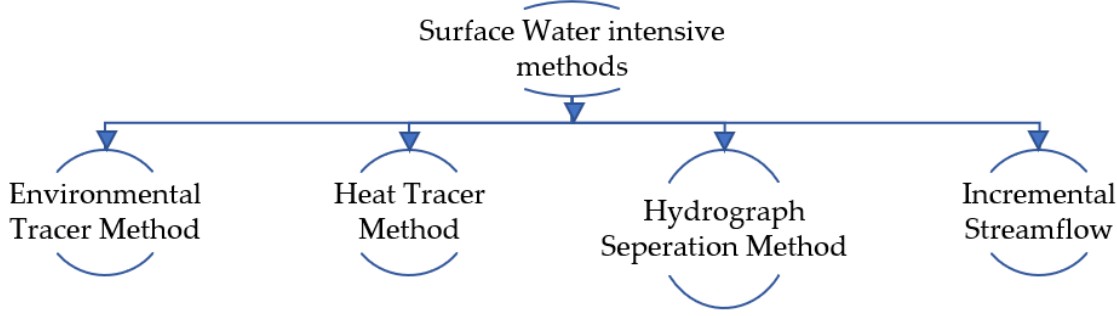

**Figure 9.** Methods for quantifying surface water interaction with groundwater.

Environmental tracers are very commonly employed in determining different phenomenon of human interest, out of which the GW–SW interactions is one. The traces generally used for this purpose are stable isotopes like deuterium ($^2$H) and oxygen-18 ($^{18}$O), along with radioactive isotopes like Radon-222 ($^{222}$R) or strontium ($^{38}$Sr) and chemical indices such as ion concentrations and electrical conductivity. To understand the precipitation-causing vapor source in a given region, the determination of the stable isotopic composition ($\delta^{18}$O and $\delta^2$H) of rainfall, surface water, and groundwater is done [56]. Katz [57] demonstrated chemical and isotopic tracers to understand the GW–SW interactions in a karst area. Further, these tracers are used for estimating the recharge of groundwater [58], determining the surface-water effusion to groundwater [59], for runoff process identification, and complex relations reflection among the river and groundwater in the alluvial plain [15]. Harvey [60] demonstrated environmental tracer approaches to be efficient and robust in quantifying and characterizing the GW–SW exchange. The tracer-based hydro-graph separation method uses isotopic and geo-chemical tracers to determine the origin of streamflow components and quantify the groundwater contribution to surface water. Specifically, the isotopes $^{18}$O and $^2$H are used to differentiate between rainfall events and pre-event flows, allowing for the identification and quantification of differences in isotope composition between rainwater and old catchment water [61].

The heat tracer method measures groundwater and surface water temperature to determine their temperature difference and assess groundwater supply to surface water or vice versa. While groundwater temperature is relatively stable year-round, surface water temperature fluctuates widely with daily and seasonal variations. Winter [1] used the heat tracer concept for distinguishing between the reaches as losing reaches have highly variable sediment and surface water temperatures, whereas gaining reaches have reasonably stable sediment temperatures and muffled diurnal changes in surface water temperatures. In addition, researchers have used this technique to characterize GW–SW interactions and quantify groundwater discharge to a surface water body using the heat balance equation.

Linsley [62] and Hornberger [61] used the Hydrograph separation method for the base flow and interflow or quick flow. Then to get the groundwater contribution to the surface water body, the base flow is extracted from the stream [63]. Graphical techniques, numerical algorithms, and many automated methods have been developed for hydrograph separation [64–66]. Many a time, it is found that the stream flow is contributed by sources like bank storage, lakes, wetlands, soils, or snowpack, then it becomes difficult to get the quantification of groundwater just by using the concept of base flow.

Incremental streamflow is an elementary method in which streamflow discharge at successive cross-sections is measured. The corresponding increment or decrement in the discharge gives an idea of the nature of the relationship between groundwater and surface water. If the discharge increases between two points of measurement, then it is understood that in that stretch of the stream, groundwater contributes to the surface water body and vice versa. To measure the streamflow discharge, various methods are used, such as the velocity gauging method in which a current meter is used [67], the use of gauging flumes [68], or the use of the dilution gauging method [69]. However, the drawback with this method is the unaccounted source or sink within the test length, so to overcome this problem, the velocity gauging method alongside the dilution gauging method was proposed [70].

### 5.1.3. Transition Zone Intensive Methods

The transition zone between groundwater and surface water is an important area for water resource management. It acts as a buffer between groundwater and surface water and helps maintain the quality and quantity of water resources. The interaction between groundwater and surface water in this zone depends on factors such as geology, topography, hydrology, and human activities. It supports diverse ecosystems, such as wetlands and riparian habitats, which rely on the exchange of water between groundwater and surface water. Understanding the various aspects of the transition zone, including

the hydrological, geochemical, ecological, and sociological factors, is crucial for effective water resource management and the preservation of ecosystems. Many researchers have studied the interaction in this zone and tried to analyze the effect of varying transition zone characteristics on the interaction [71–73].

Transition zone flow is analogous to flow in the aquifer, and here we are mainly concerned with estimating elements of the Darcy equation (Figure 10). To obtain these components, we will use the same methods used for the aquifer. Hence, in this section, we will discuss methods particular to the transition zone in detail.

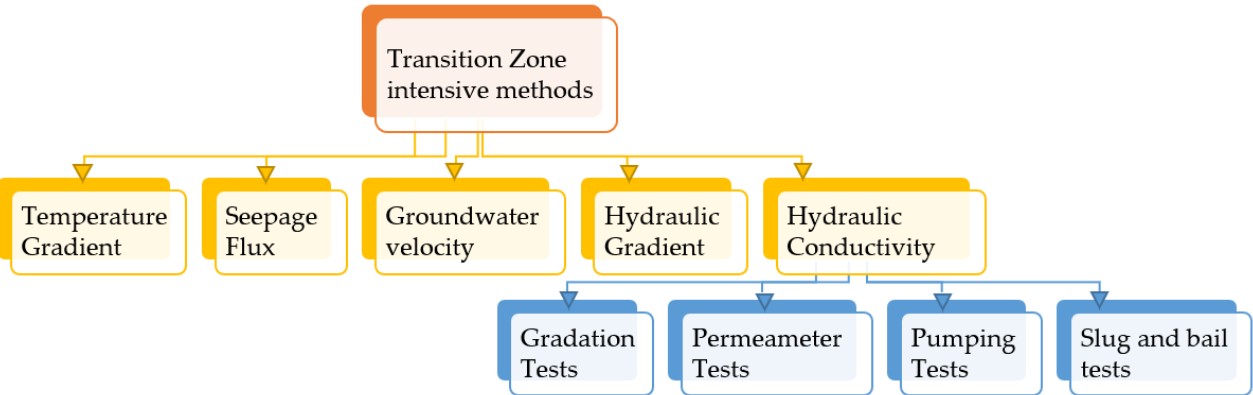

**Figure 10.** Methods for quantifying transition zone water interaction with groundwater.

As discussed earlier, the Temperature Gradient method uses the difference in temperature to understand how the interaction is taking place. As reported by Domenico and Schwartz [74], heat flows through these zones is governed by the heat transport equation. Similarly, Silliman [75] has suggested different analytical and numerical solutions which are further used to get the seepage flow beneath the stream.

Seepage meters are a low-cost method for directly measuring seepage flux between groundwater and surface water bodies. By combining flow observations from seepage meters with hydraulic head readings from mini-piezometers, the hydraulic conductivity of bed sediments can be determined. Lee [76] proposed a bag-type seepage meter consisting of a bottomless cylinder vented to a deflated plastic bag. The bag-type seepage meters have been widely utilized in lakes, estuaries, streams, etc. [77,78]. Recently developed are the automated seepage meters, which can be used in a difficult location and with high precision without the need for human presence at the site. They are first calibrated and then installed on the site to get time series data with high temporal resolution [79]. In the case of streams, due to flowing water, the collection bag may distort or fold, affecting the hydraulic head in the bag, and as a result, the seepage meter detects a drop or increase in flux, so multiple readings should be taken to get reliable data [80].

*5.2. Analytical Modeling Methods*

The accurate assessment of flow paths in the hyporheic zone requires considering all three components of groundwater velocity underneath and along the riverbed. Simplifying this complex flow involves modeling horizontal and vertical flow separately, and coupling them using a Darcy-type linear formula. This considers the difference in groundwater levels in the river and neighboring aquifers, assuming inverse proportionality to the resistance of the sediment layer of the riverbed. [5,81]. Most numerical models of regional groundwater flow depict water exchange within the river–aquifer systems as solely vertical water seepage through riverbed sediments, while the Darcy-type model (DM) approximates the corresponding water flux (per unit length of river stretch segment) [82]. Researchers typically use a 2D approximation to solve the problem of groundwater and surface water interaction. They consider a middle vertical plane of a homogeneous section of a river length for calculation purposes and neglect the side banks of the river. However, this

approximation is only valid when the riverbanks are much smaller than the width of the riverbed. If the riverbank has significant dimensions, the side banks will contribute considerably to the exchange process, rendering the approximation invalid.

### 5.3. Numerical Modeling Methods

A numerical model is a mathematical tool that simulates a phenomenon using differential equations and boundary conditions and requires initial conditions for transient simulations [83]. Groundwater flow models investigate and forecast groundwater system behavior with different types, including steady, one-dimensional, two-dimensional, and quasi-three-dimensional models [84]. The model choice depends on input data, aquifer type, and hydrogeological system, with complex groundwater systems receiving attention from researchers in the development of numerical models to simulate GW–SW interactions.

In the case of surface water–groundwater interactions, coupling different models is generally done as a single model may not be efficient enough. Hence, water has been evaluated using hydrological models. Further, these hydrologic models have been categorized into three categories depending on the hydrologic component they simulate. These are as listed below:

(i)     Runoff simulations using hydrological models;
(ii)    Simulations of groundwater systems using hydrogeological models;
(iii)   Physical properties based ISSHM (Integrated Surface–Subsurface Hydrological Models).

As discussed, there are mainly three models for modeling the interaction between surface and groundwater. Each type of model has its limitations in terms of applicability. The hydrological models are proficient in modeling surface water phenomena, while the hydrogeological models provide dependable results for modeling the groundwater processes [85]. However, modeling the interaction includes complex processes, hence any one of these models is found to be unsuccessful. Several variables impact the interaction of groundwater and surface water, like hydro-climatic factors, physiographic structure, groundwater and surface water head difference in the catchment, and geometry of flow within the aquifer [86]. This brings the coupled models into the picture. For a larger scale, such as a regional scale, the best-accepted models are the fully coupled models like CATHY, and MIKESHE if all hydrological processes are considered. The advantage of these models is that they can accurately model the groundwater and surface water interactions using process-based equations in conjunction with 3D subsurface demonstrations [85]. For the local scale modeling, the loosely coupled models such as SWAT-MODFLOW, MODFLOW-MODHMS [87], and MODBRANCH are found to be more accurate and less time-consuming [84]. The loosely coupled scheme has the advantage of flexibility; individual tools can be applied to each process within a particular environment rather than one tool for all processes [65]. In addition, these models also provide the option to choose the preferable platform during different stages of the modeling [76]. Some common models used for the estimation of GW–SW interaction are listed in Table 1.

**Table 1.** Some common models used for GW–SW interaction estimation.

| Model Name | Discretization/Equations Solved | | Applications |
|---|---|---|---|
| | **For GW** | **For SW** | |
| CATHY | 3D Finite Element | 1D Finite Difference | • 3D sub-surface flow in saturated porous media<br>• Surface routing on hill slopes and stream channels |
| MODFLOW-MODHMS | MODFLOW (3D Finite Difference) | 1D Saint Venant | • Watershed management<br>• Wetland management<br>• Contaminant transport |

**Table 1.** *Cont.*

| Model Name | Discretization/Equations Solved | | Applications |
|---|---|---|---|
| | **For GW** | **For SW** | |
| MODBRANCH | MODFLOW (3D Finite Difference) | BRANCH (1D Saint-Venant) | • Wetland management<br>• High backwatering effect Canal networks<br>• Surface water changes rapidly affect groundwater |
| MIKE SHE | 3D Finite Difference | 2D Saint-Venant | • Floodplain management<br>• Groundwater-induced flooding<br>• Nutrient transport and fate<br>• Wetland restoration |
| SWAT-MODFLOW | MODFLOW (3D Finite Difference) | SWAT | • Climate change impact on hydrological processes<br>• Aquifer Evapotranspiration |

*5.4. Semi-Analytical Methods*

The accuracy of the numerical models is highly dependent upon the resolution and meshing of the concern process. Many a time, the models are unable to take into account the free boundary problems, variation of the head, etc. [88], which in turn causes the properties of the domains to be altered or modified, causing the results to deviate from the actual scenario [89,90]. The same can be seen in the case of a surface water body in which it has been reported that the underlying sediment often needs to have accurately meshed as the practical mesh size is larger enough to ignore such minute features. For such a case, a 1-D approximation, such as incorporating a river coefficient, is used to include the effect of the sediment [90].

In order to address these problems, grid-free semi-analytical methods can be used, which provide the benefits of both analytical and numerical modeling methods. These methods mainly consist of Hankel, Laplace, Fourier transforms, series solutions, etc. Many researchers such as Craig [91], Mishra and Neuman [92], Mishra et al. [93], Tartakovsky and Neuman [94], Tristscher et al. [95], and Wong and Craig [96] have used these methods in conjunction with numerical inversion or Weighted Least Squares (WLS) to solve the mathematically and geometrically complex problems. In groundwater–surface water interactions, these methods provide exact solutions to the governing differential equations for linear or linearized problems while a very accurate solution when the process is considered in 2-D and 3-D [97].

Ward and Lough [98] modeled 2-D groundwater–surface water interaction for a domain with simple boundary conditions using Laplace-Fourier double transform method. Wong and Craig [96] have used semi-analytical methods to model more complex processes like multi-layer aquifers and heterogeneity. These methods are also extended to model 2-D steady state saturated-unsaturated free boundary [95], but more needs to be contributed for 3-D modeling of stratified unconfined aquifers for saturated and saturated-unsaturated steady flow.

**6. Discussion and Applicability of the Methods**

A large number of literature was referred to develop this work. The works that were referred to are tabulated in Table 2 under different methods of quantification of interaction.

Darcy's approach is one of the most common approaches whose measuring aspects have already been discussed in detail. This method requires data collection from the site, and mainly in situ, hydraulic conductivity and hydraulic gradient are measured. Many

researchers have based their studies on the outcomes of this method, which has been used to understand the interactions in various sites worldwide. Most of the studies used piezometers [4,99,100] and mini-piezometers [101,102] to obtain hydraulic gradients, and by quantifying the hydraulic gradient, they understood the nature and direction of flow. Hence, it can be thought of as an easy process to be executed by installing piezometers at the stream bed and obtaining their levels at different times to conclude the phenomenon. Further, a few works also considered obtaining the hydraulic conductivity through slug tests [101], which would refine the study as information on stream bed properties would help understand the exchange process better. Many studies also used groundwater level maps to generate the hydraulic head data and obtain the hydraulic head from it [103–105]. The method was found to be widely popular among researchers across the world, with applications distributed evenly in different countries like Ethiopia, Italy, Denmark, England, China, and more. Menció [106] studied the relationship between the aquifer and stream in the Onyar River watershed, using Darcy's law and Mass balance methods. They found that the stream gained water from the aquifer from February to May but acted like a losing river from July to October. Cremeans [101] compared four tools for quantifying the exchange of groundwater and surface water in Grindsted Å, Denmark. They found that streambed point velocity probes (SBPVPs) was the best method for sandy riverbeds, while seepage meters and temperature profilers were useful for estimating flow trends.

Hydrochemistry can estimate GW–SW interactions using tracers such as stable isotopes, radioactive isotopes like radon and uranium, chloride or alkalinity, and electrical conductivity. This method assumes an equal concentration of tracers in GW and SW and complete mixing between the sources. Stable isotopes estimate GW flow patterns, age, residence times, and evaporation effects on GW and SW. It was seen that many works [4] [107,108] were done with the stable isotopes $H^2$ and $O^{18}$. $O^{18}$ and $H^2$ are the isotopes present in the rainfall; hence, if they get infiltrated into the groundwater, they can be tracked to understand the recharge phenomenon. Besides this, Radon ($^{222}$Rn) was also found to be used extensively [103,109,110] because this element is said to dissolve when in contact with the atmosphere like surface water but in groundwater, it is an exponentially growing element hence the interaction, can be estimated. In addition to these, elements of normal water chemistry like $Na^+$, $Ca^{2+}$, $Mg^{2+}$, $HCO^-$, and $Cl^-$ have been a way to understand the interaction and are found to be used in conjunction with other methods many a time [107,109,111]. These works also used electrical conductivity in conjunction with these ions to understand the nature of the interaction. Uranium isotopes were also used in a few works [111,112]. Freitas [107] conducted hydrological dynamics experiments at the Pantanal wetland in Brazil, finding that alkaline-saline lakes had higher hydraulic heads than groundwater and groundwater near the lakes had high isotopic concentrations. Hence this approach is found to be self-sufficient to fully estimate the process using one or more tools from the same domain, and it is the most widely used technique for this estimation

The water budget approaches are the mass balance method, where the flux in the channel is measured at different fixed locations along the flow. Then the increment or decrement is directly related to the gaining or losing stream accordingly with the condition that all other extractions and additions are taken care of. Different works have been done using this technique for the estimation of interaction [99,113,114]. However, unlike the previous approach, this approach may readily incur errors. The measurements of the discharges should be taken at low discharge places, or measurement errors can easily take place. Most of the studies [100,109] often utilize multiple approaches to investigate small-scale groundwater–surface water connectivity, such as obtaining accurate results requires closely spaced flow gauging and extensive piezometric surveys, which can be time-consuming and prone to errors. Larned [114] used flow gauging data to determine the gaining and losing reaches of the Selwyn River in New Zealand, finding similarities in solute concentrations in both groundwater and surface water sources. Longa [115] evaluated the riverbed conductance of the Saigon River Basin using piezometric head and field seepage measurements, establishing the water balance of river-groundwater using

stable isotope samples of groundwater, river, and precipitation. Li et al. [116] used the cumulative exchange fluxes method to study the Taizi River Basin of China, observing changes in water levels at different sections. This method is proven to be more beneficial for large-scale interaction estimation. Figure 11 depicts the popularity of this method showing its percentage share in the total reviewed works.

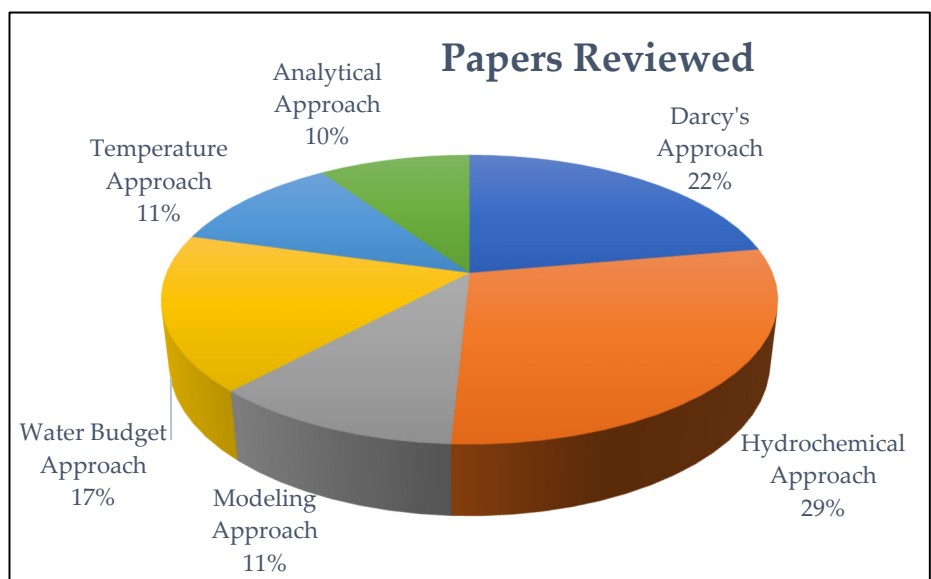

**Figure 11.** Share of the studies reviewed under each type of approach.

The temperature approach estimates the interaction between surface water and groundwater temperature. In addition to this, many heat tracers can be used to quantify the interaction. Temperature can be measured in streams or wells by using probes, sensors, and data loggers, and further vertical and horizontal temperature profiles can also be measured by using sensors. Temperature depth profiles are built to find the stream bed flux. T-arrays were also found to be successfully producing the time series [103] with the help of a temperature logger fitted in the arrangement. Many of the works were found to be done paired with some other approaches, Passadore [117] used the Darcy approach along with the heat tracer method in which thermal monitoring of fluxes through the river bed was done and verified with Darcy's outcomes. Thomas [118] established a relation between the TDS and DO with temperature along the river profile using a multimeter. Coluccio [102] used a tool named VFLUX to analyze the temperature signals and understand the phenomenon taking place through the streambed. However, it was difficult to quantify the groundwater and hyporheic flow through this method precisely, so he used Darcy's and hydrochemistry in conjunction. So, this can be inferred since this technique can give dependable results when used with other parallel approaches.

Modeling approaches are now an essential method for estimating groundwater–surface water interactions due to their usability at different scales and ability for future prediction. These models range from simple conceptual models to complex models that take all factors into account and simulate the real physical state for higher precision. A transient MODFLOW model was calibrated using targeted field observations and "soft" information from local water authority experts to estimate the process. MODFLOW was found to be used to estimate the process after calibrating the model with the local data collected from the concerned site [119]. Further, a 3D groundwater flow MODFLOW model coupled with a 1D analytical solution to the heat transfer equation was employed to understand the vertical flux of low-lying lands to see that even though vertical fluxes were estimated nearly the same by both, an overall estimate of MODFLOW was twice that of the 1D analytical solution [120] which further proved the applicability of the method to estimate the interaction. In addition, works by coupling MODFLOW with other models

were often done, like SWAT-MODFLOW, to combine the calibration of both surface water and groundwater parameters so that improved results can be obtained for the processes. Similarly, a linked SW–GW modeling approach like MODFLOW along with SWAT grid was used [103] to obtain better rainfall-runoff modeling, which further helped to precisely simulate runoff in droughts and dry seasons. MIKE SHE, a hydrologic model, was combined with MIKE 11, a hydrodynamic model, to model the complex hydrologic and hydraulic interactions of lowland river basins and was proved to be very reliable for the present estimation and prediction [121]. Few works used a single modeling software like GSFLOW directly [122] for the process estimation. Reeves and Hatch [123] used heat as a tracer to understand groundwater and surface water interaction by creating a 3D flow model and comparing it with the 1D heat tracer method. They found that both effective thermal diffusivity and temperature-derived flux were prone to errors as the flow became non-uniform and non-vertical in the case of groundwater flow. Hence, numerical modeling approaches are reliable and can accurately estimate processes without requiring parallel techniques, according to research. The approach can replace tedious site investigations if the model is calibrated with correct data and proper physical simulations are developed.

**Table 2.** Research works referred to, for each approach of GW–SW interaction.

| Sr. No. | Method Applied | Fundamental Mechanism | Related Studies |
|---|---|---|---|
| 1. | Darcy approach | Measure components of Darcy Law experimentally and use Darcy's Law | Kebede et al. (2021) [100], Sadat-Noori et al. (2021) [103], Larned et al. (2015) [114], Doering et al. (2013) [99], Banks et al. (2009) [4], Coluccio (2018) [102], Burbery and Ritson (2010) [113], Menció et al. (2014) [106], Acuña and Tockner (2009) [109], Cremeans et al. (2020) [101], Freitas et al. (2019) [107]. |
| 2. | Hydrochemistry | Quantify different chemical ions in the samples of groundwater and surface water to understand their extent of interaction | Longa and Koontanakulvong (2020) [115], Kebede et al. (2021) [100], Sadat-Noori et al. (2021) [103], Larned et al. (2015) [114], Doering et al. (2013) [99], Banks et al. (2009) [4], Coluccio (2018) [102], Burbery and Ritson (2010) [113], Guggenmos et al. (2011) [111], Acuña and Tockner (2009) [109], Navarro-Martínez et al. (2020) [112], Carol et al. (2020) [108], Ferreira et al. (2018) [110], Freitas et al. (2019) [107]. |
| 3. | Heat (Temperature) approach | The difference in the temperature between groundwater and surface water | Thomas (2021) [118], Sadat-Noori et al. (2021) [103], Doering et al. (2013) [99], Banks et al. (2009) [4], Coluccio (2018) [102], Acuña and Tockner (2009) [109], Passadore et al. (2015) [117]. |
| 4. | Numerical modeling | Replicate the actual scenario into a software environment with some inputs taken through experimentations e.g., hydraulic conductivity | Tran et al. (2020) [122], Deb et al. (2019) [104], Waseem et al. (2020) [121], Ghysels et al. (2021) [120]. |
| 5. | Water Budget | Any increase or decrease in the quantity or quality of surface water is due to its source, which is the groundwater | Kebede et al. (2021) [100], Li et al. (2020) [116], Larned et al. (2015) [114], Doering et al. (2013) [99], Banks et al. (2009) [4], Burbery and Ritson (2010) [113], Guggenmos et al. (2011) [111], Menció et al. (2014) [106], Acuña and Tockner (2009) [109] |
| 6. | Analytical/semi-analytical modeling and other approaches | Use equations of groundwater flow and different conditions to get nearer to the actual phenomenon | Ghysels et al. (2021) [120], El-Rawy et al. (2020) [124], Keery et al. (2006) [125], Thomle et al. (2020) [126], Johnson (2012) [119], Nawalany et al. (2020) [82]. |

The analytical approach is found to be useful in investigating the interaction between groundwater and surface water flow, especially for streams and adjoining areas. They are simple to be implement and solve with respect to modeling approaches where many data are required, and assumptions are made to get things in line. Many works have been published relative to this approach, such as that of Nawalany [82], where the validity of the Analytical Hyporheic Flux approach (AHF) formulated using the exchange flux from both the streambed and the river banks was assessed. Similarly, El Rawy [124] used the Stream River Ecosystem (STRIVE) package for analyzing analytically the one-dimensional and two-dimensional confined as well as unconfined interactions. However, both of these works were accompanied by numerical models. Nawalany [82] used the SEEP2D and Darcy model for validation, while the latter used MODFLOW for the same purpose. For both cases, it was observed that they agreed with the model outputs, and hence their applicability was confirmed. Further, Keery [125] developed an analytical method based on the temperature time series data from the site and applied it to estimate fluxes through both sources. Besides all these approaches, many new methods were also used, such as Johnson [119], who used the time-series and time-frequency analysis of 3D transient electrical resistivity changes to monitor groundwater–surface water interaction. Similarly, Thomle [126] developed a probe to obtain the in situ porosity linking to the Darcy flux, as discussed before. Natural tracers were used to verify the probe's accuracy, and it was fitted with pressure sensors and temperature sensors. The probe showed very good accuracy for downward and upward flow velocity measurements.

The advantages and disadvantages of each estimation method are tabulated in Table 3. This helps us choose the correct approach for our study in particular. Furthermore, this, to obtain the most accurate and dependable output from the investigation of GW–SW, we have to consider two significant steps: (a) Choosing the scale to be considered for measurement and (b) Choosing the appropriate method for estimation of the process.

**Table 3.** Pros and cons of different approaches of GW–SW interaction.

| Approach | Advantages | Disadvantages |
|---|---|---|
| Darcy approach | • Applicable to both small- and large-scale studies and also appropriate for heterogeneous reaches.<br>• Existing wells can be used to save time and expense.<br>• Piezometers may be used instead of wells which are easy and quick to install with variety for serving a different purpose. | • Hydraulic conductivity, the important factor of this method, generally varies spatially hence needing a lot and continuous measurements in the site.<br>• New wells are expensive to install<br>• Simultaneous measurements from the instruments are required for accuracy. |
| Numerical modeling approach | • Can be used to predict the physical processes as well as quantify the present phenomenon.<br>• Can be used for very complex phenomena by modeling accurate site circumstances<br>• Many model packages are available, which can make work easier and faster with more accurate results<br>• Combined modeling can also be done to enhance the outputs depending on the nature of the field and work | • Proper inputs to the model about field conditions may not be available, always causing the error to an outcome.<br>• May prove to be costly if better precision is required as field data will have to be collected using various instruments and methods<br>• High time and computation may be required for complex process modeling |

**Table 3.** *Cont.*

| Approach | Advantages | Disadvantages |
|---|---|---|
| Hydrochemistry (Use of isotopes) | • Large-scale interactions are estimated better<br>• Best for those reaches where isotope concentrations between the two sources are high enough<br>• Can be estimated relatively easily and with higher accuracy in the lab | • Isotope concentration may vary with time and space<br>• Not much useful if the concentration of isotope is not high for the difference to be quantified<br>• The nature of tracers may affect the quantification and sometimes even the biogeochemical processes of water |
| Temperature studies | • Discharge and recharge reach can be differentiated along with the quantification of flux water<br>• Faster data collection if sensors and probes are used<br>• Depending upon the instrument used, the data can be easily and accurately available | • Can prove to be costly depending upon the method used to measure the data<br>• Continuous data needs to be measured at variety of locations<br>• Temperature difference between both sources should be high enough to be quantified |
| Water Budget | • Relatively simple to calculate<br>• More accurate for large-scale studies and homogenous aquifers<br>• Useful to chalk out the river loss and gain reaches | • Not suited for heterogeneous aquifers and small hyporheic reaches<br>• Every input and output site must be accurately measured simultaneously<br>• Can prove to be costly if more places of quantification are there<br>• Streambed conductance is not taken into account |
| Analytical/semi-analytical and other approaches | • Simple to be applied<br>• Easier to be implemented than other approaches<br>• Not a site-specific approach, generalized to all and can be applied to other scenarios with slight modification<br>• Developed mostly for both confined and unconfined cases | • Site characteristics may not be properly incorporated as it is a general approach<br>• Errors may boom if data and assumptions are not accurate to the site |

(a)　Choosing the scale to be considered for measurement

The performance of any estimation method depends on the quality of the data used. The best scale to use for a given study depends on the extent of the area or region being studied [127]. Point estimates are suitable for site-specific studies, while large-scale estimates are preferable for regional studies [85]. Time scales can also be relevant in some cases. When tracking the flow of a dye or plume through groundwater, closely spaced wells are necessary to avoid missing the plume's path [128]. Point measurements are usually taken using seepage meters, piezometers, grain size analyses, and permeameter tests, while isotope-related tests like tracer tests and pumping tests are larger in scale. Scale measurement is especially important for heterogeneous media, where small-scale measurements are preferred [129]. Finally, correct model inputs, whether spatial or temporal, require special attention to ensure accurate outputs. Instruments like probes and seepage meters are useful for measuring temperature and water volume data, respectively, over a long temporal span. Additionally, data loggers can provide extended temporal data, while other instruments provide measured parameter data at a specific point [129].

(b) Choosing the appropriate method for estimation of the process

The choice of method for estimating groundwater and surface water depends on the requirements of the study. The general advantages and disadvantages of different

methods are summarized in Table 3. Mapping techniques and heat tracers can provide an overview of the process, while the Darcy approach or mass balance methods can quantify the process [13]. Pumping tests may be used for contaminant transport, and temperature profiles, piezometer methods, and slug tests may be used to monitor highly variable parameters. The selected method should be checked for its assumptions and applicability in the current case, along with the availability of required tools at the site [42]. Multiple methods should be used in conjunction to rule out errors related to scale and the method itself. Analytical methods can be considered to get the best results [100].

## 7. Conclusions and the Way Forward

The phenomenon of the GW–SW interaction is summed up here by addressing all the salient features of the process. The important conclusions drawn from the study are:

- The interaction mechanisms were shown to be dependent on base flows and interflows in general, along with flows like translatory, macropore, groundwater ridging, and return flow for a quick response after a storm event. Alongside this, the dependency of the interaction process on the scale of interaction was explored as an overview of the local, intermediate, and regional flow systems.
- It was observed that the large-scale interactions explained how seepage from the channel bed would take place in the absence of a continuous groundwater flow system boundary. The hyporheic zone dependency on stream bed materials' hydraulic conductivity and bed topography was proven.
- The methods of interaction and instruments used were found to be dependent on the scale in which the process needs to be quantified. Aquifer zone methods used Darcy's Law as the basis of estimation, but due to the variability of different parameters used in this method, it was found highly prone to errors.
- For surface water, there are several methods like the Environmental Tracer Method, Heat Tracer Method, Hydrograph Separation Method, and Incremental Streamflow. Of these, the Environmental Tracer Method was found to be the most widely and successfully used technique for interaction estimation. In the transition zone, most of the measuring parameters resembled that of surface water.
- Taking into account the strengths of analytical and numerical modeling techniques, 2-D and 3-D semi-analytical solutions can be used to simulate the complex physical and mathematical phenomenon of groundwater–surface water interaction. These methods are efficient even to solve and simulate those typical interaction problems without the use of grids as at many locations meshing cause the solution to deviate from the actual process taking place.
- The most important factors in achieving an accurate interaction quantification are the scales and methods chosen. It is always better to get results through both large-scale and small-scale measurements since adhering to a single scale may ignore the intricacies that are detectable in the other. Further, the methods are suggested to be used in conjunction with another technique to avoid any error in the result.

Works in the existing literature have focused mostly on experimental methods for studying hyporheic zone flow processes. However, there is potential and scope for the use of analytical methods or numerical modeling in conjunction with these methods. The hyporheic zone flow processes are complex and significant for ecology and contaminant transport, but there is difficulty in differentiating between hyporheic exchange and groundwater discharge, leading to errors in quantification. This sector can be made an area of focus in future ventures. Adequate knowledge of the corresponding interactions would be greatly beneficial in developing methods to estimate groundwater and surface water interaction, and preventing damage to the water system and loss of human life.

**Funding:** This research received no external funding.

**Data Availability Statement:** Not applicable.

**Conflicts of Interest:** The authors declare no conflict of interest.

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
