# Peer review of "A Review on the Research Advances in Groundwater–Surface Water Interaction with an Overview of the Phenomenon"

_water, doi:10.3390/w15081552_

Round 1

Reviewer 1 Report

Dear Authors,

the topic of your paper is potentially relevant.

However, your paper at present is more in the form of a draft and very long. I suggest you reduce it to about 600 lines up the conclusions section.

So, additional work is needed, before it may be considered for publication.

In particular, there are several sections where text does not follow a coherent flow.

The introduction is too long.

Line 32-47: this is a trivial section. Please sum up in 3 lines.

Your paper can start at line 57

Line 78-96: please sum, this is a paper not a thesys.

Lines 106-112. on what search do you base this text? or are you citing existing works?

In the text you go back and forth on the hyphoreic zone: please sum up

Line 121: this is false - recongniction of pumping influencing instream flows was much earlier.

Line 138-152 this section is not organised

Section 2 lines 195-205 are still for the intro section.

Other comments:

- you should detail the methods used to prepare this review;

- in the introduction please present the main topics discussed in the paper and the flow of reasoning;

- the relevant topic of indeuced riverbank flow is not dealt with. Please see:

Rossetto, R.; Barbagli, A.; De Filippis, G.; Marchina, C.; Vienken, T.; Mazzanti, G. Importance of the Induced Recharge Term in Riverbank Filtration: Hydrodynamics, Hydrochemical, and Numerical Modelling Investigations. Hydrology 20207, 96. https://doi.org/10.3390/hydrology7040096

- section five should be named Methods for analysing ...

- why transition zone intensive methods?

- modelling method - any modelling method not able to simulate gw/sw exchanges is not fit for purposes.  Why you mention MODHMS? also the simpler MODFLOW-2005 is able to capture gw/sw exchanges.

This section has to be completely revised.

- do not understand why you have sect 6 and then 6.1 . This is rather chaotic. Please shortn up the disucssion section.

I am not able to detect plagiarism without having an anti-plagiarism software.

Reviewer 2 Report

I read the text with pleasure that someone took the trouble to collect information about GW-SW interaction in one article. For me, the text is clear and interestingly leads through the history of research on this issue.

I think that the enrichment of the introduction with the SW-GW interaction measurement methods would enrich the whole text. There are several methods of measuring or modelling GW-SW interaction:  for example DOI: 10.24425/aep.2021.136450; https://doi.org/10.5194/hess-16-2329-2012 DOI:10.7717/peerj.13418; and others can be given.

Round 2

Reviewer 1 Report

Dear Authors,

this is a too long paper for my liking, but I see you are not really willing to reduce its lenght.

So, this is in the hand of the academic editor handling the paper.

I only suggest that a carefull read is given to this paper before it's published because of the high risk of clerical errors or poorly readable sentences - which is of course a matter of editing.

Also, I am not able to check for plagiarism - internally the editorial staff should make use of a software for such purpose.

Regards
